# Estimation Method of an Electrical Equivalent Circuit for Sonar Transducer Impedance Characteristic of Multiple Resonance

**DOI:** 10.3390/s23146636

**Published:** 2023-07-24

**Authors:** Jejin Jang, Jaehyuk Choi, Donghun Lee, Hyungsoo Mok

**Affiliations:** 1Research and Development Division, Tin Technology Co., Ltd., Seongnam 13212, Republic of Korea; jjjang@tin-technology.com (J.J.); jhchoi@tin-technology.com (J.C.); 2Agency for Defense Development, Changwon 51678, Republic of Korea; leedhun@add.re.kr; 3Department of Electrical Engineering, Konkuk University, Seoul 05029, Republic of Korea

**Keywords:** sound navigation and ranging, particle swarm optimization, electrical equivalent circuit, multiple resonant characteristics

## Abstract

Improving the operational efficiency and optimizing the design of sound navigation and ranging (sonar) systems require accurate electrical equivalent models within the operating frequency range. The power conversion system within the sonar system increases power efficiency through impedance-matching circuits. Impedance matching is used to enhance the power transmission efficiency of the sonar system. Therefore, to increase the efficiency of the sonar system, an electrical-matching circuit is employed, and this necessitates an accurate equivalent circuit for the sonar transducer within the operating frequency range. In conventional equivalent circuit derivation methods, errors occur because they utilize the same number of RLC branches as the resonant frequency of the sonar transducer, based on its physical properties. Hence, this paper proposes an algorithm for deriving an equivalent circuit independent of resonance by employing multiple electrical components and particle swarm optimization (PSO). A comparative verification was also performed between the proposed and existing approaches using the Butterworth–van Dyke (BVD) model, which is a method for deriving electrical equivalent circuits.

## 1. Introduction

Sound navigation and ranging (sonar) systems detect underwater objects by utilizing electrical energy-to-sound energy conversion. Sonar power systems consist of the following components, as shown in Figure 1 [1,2,3]: (1) a direct current (DC) power supply, which serves as the electrical energy source required for generating acoustic energy; (2) a power converter, which converts the DC voltage into alternating current (AC) to provide the desired electrical energy supply for the required sound signal intensity; (3) an LC filter (or low-pass filter), which removes unnecessary frequencies for acoustic detection; (4) an impedance-matching transformer, which eliminates the reactive power generated by the material characteristics of the sonar sensor; and (5) a sonar sensor, which converts the input electrical signal into an acoustic signal.

Sonar systems have been developed for precise and wide-ranging exploration aimed at broadband operation [4,5], with high capacity, and high efficiency [6]. When designing high-power and high-efficiency power converters for sonar systems, reactive power and active power must be minimized and converted into acoustic energy [6]. After designing a sonar power system that meets these requirements, it is essential to verify the sonar sensor’s performance. However, the electrical characteristics of sonar sensors can vary depending on the installation environment, such as terrestrial or submarine, the sensor type, and the sensor array structure. Therefore, while designing and validating high-power and high-efficiency power converters, accurate electrical equivalent models of the sonar sensor are required. Equivalent circuit models can be of various types and reflect the physical characteristics or operational features of the sonar sensors.

The equivalent circuit model originated from Mason’s one-dimensional (1D) model of a piezoelectric transducer proposed in 1942 [7]. In this model, the physical movement of the piezo-ceramic was compared with that of a spring, and an ideal transformer was used to represent the electrical circuit. In 1961, Redwood proposed a method that used electrical transmission lines to model the transient behavior of sonar sensors [7]. In 1970, the Krimtholz, Leedom, and Mattthaei (KLM) model was proposed to simulate sonar sensors operating in the high-frequency range [7,8,9,10,11,12]. In 1994, Leach proposed the Butterworth–van Dyke (BVD) model as a replacement for the conventional Mason model and provided the simplest approach for simulating multilayered sensor structures. Although this methodology is not well suited for high-frequency characteristics and multiple resonances [7], its application as a modeling circuit remains predominant owing to its remarkable accuracy at the resonance points. 

An analytical method using approximation techniques in a certain range of sonar sensor impedances was employed to estimate the parameters of equivalent circuits [7,13]. However, this approach is ineffective when the impedance variation at the resonant frequencies is small. To overcome this limitation, an alternative approach that calculates the equivalent circuit parameters based on the resonance frequencies within the actual impedance of the sensor was proposed. However, this method yielded inaccurate results when deriving the parameters of an equivalent circuit [14]. Building on this calculation method, Ramesh and Ebenezer [15] proposed a parameter estimation technique using the least squares method. They assumed that each resonance point is independent and treated the equivalent circuit for each resonance as a parallel RLC branch. They essentially proposed a method for deriving an equivalent circuit and its parameters for a sensor with two resonance frequencies [15]. A drawback of this method is that the estimation errors increase when the resonance frequencies are close to each other. Consequently, to accurately model sensors with abrupt impedance variations is challenging. To address these limitations, recent studies have proposed parameter estimation approaches using particle swarm optimization (PSO) [16]. However, owing to the limitations of the frequency degrees of freedom of the equivalent circuit, even using such methods, the modeling of rapidly changing impedance characteristics remains difficult. Therefore, in this study, we devised an electrical equivalent circuit derivation approach by constructing an equivalent circuit composed of multiple RLC components configured in series and parallel. The PSO algorithm was employed to determine the physically realizable values of the equivalent circuit component parameters. To validate the proposed approach, electrical equivalent circuits of sonar sensors with single, dual, and multiple resonances were derived using both conventional and proposed methods. The accuracy of the impedance estimation was evaluated with respect to the change in the number of resonances. 

## 2. Electrical Equivalent Circuit for Sonar Sensors

### 2.1. Conventional Electrical Equivalent Circuits

The BVD equivalent circuit, shown in Figure 2, is composed of an RLC branch consisting of resistance ($R_{1}$), inductance ($L_{2}$), and capacitance ($C_{3}$) to mimic the electrical resonance frequency characteristics, along with a parallel capacitor ($C_{0}$) representing the electrical capacitance properties of the piezoelectric component. Therefore, in this study, to reduce sensor errors caused by physical and material factors, an enhanced equivalent circuit with increased electrical resonance modes and RLC branches was employed within the BVD model. 

The electrical admittance of Figure 2 is determined as follows:(1)$$Y=\frac{I}{V}=j\omega C_{0}+\sum_{i=1}^{n}\frac{1}{R_{i}+j\omega L_{i}+\frac{1}{j\omega C_{i}}}$$

### 2.2. Proposed Electrical Equivalent Circuits

The equivalent electrical circuit in Figure 2 represents the resonances within the operating frequency range by using an RLC branch for the nth resonance. However, such equivalent models face challenges with regard to selecting different models based on the observed resonances in the measured sonar impedance data and estimating the parameters within the selected model. 

Furthermore, based on the measured characteristics of the sonar sensor, an equivalent circuit was determined. When using a large number of components, parameter estimation required a relatively long time, resulting in more accurate results. Conversely, for a small number of components, the estimation time was reduced, although the accuracy decreased, thus exhibiting a trade-off relationship. Additionally, after deriving the corresponding parameters, a trial-and-error process was performed under human intervention for additional calibration. Therefore, in this paper, we propose a high-degree model composed of 54 RLC components, as depicted in Figure 3, which allows for the selection of different equivalent circuits for each resonance frequency. Thus, the proposed model seeks to overcome the conventional limitations of the sensor characteristic simulation due to the constraints on the number of RLC components.

## 3. Estimation of Electrical Equivalent Circuit Using Particle Swarm Optimization

To derive the electrical equivalent model of the sonar sensor, the electrical characteristics of the sensor must be analyzed and an appropriate type of equivalent model must be selected. Once the equivalent model is determined, optimization algorithms are used with the measured impedance magnitude and phase characteristics of the actual sonar sensor in its operating frequency range as reference values. This allows for the extraction of parameter values at the point where the combination of parameter values minimizes errors in the impedance magnitude and phase.

As shown in Table 1, the least squares method, genetic algorithm (GA) and PSO have their own characteristics. The least squares method faces the difficulty of achieving optimal results in the presence of large errors. Meanwhile, the GA suffers from the lack of diversity among individuals, leading to convergence to nonoptimal solutions. 

In this study, the PSO algorithm was employed to derive the equivalent circuit parameters of a sonar sensor [17]. The PSO algorithm is based on swarm intelligence, which is inspired by the collective behavior of flocks of birds and schools of fish. It has the advantage of obtaining results quickly because it only transmits the best global optimum information instead of requiring overlap or mutation operations [18]. However, the PSO algorithm may suffer from convergence to the local optima rather than the global optimum if the speed and direction of the particles are inaccurate. To address this issue, the inertia weight (w) in the PSO algorithm was examined so as to improve its performance [19].

### 3.1. Conventional Method

The electrical equivalent model of the sonar sensor was derived using the PSO algorithm as shown in Figure 4, and related variables are shown in Table 2.
(1)Initialization phase: The necessary variables for the PSO algorithm are initialized, and the initial values are set. (2)Fitness calculation of the particles: The impedance characteristics of the sensor are calculated based on the estimated parameter values of the equivalent circuit and the variables obtained from the previous step. The calculated and measured impedance characteristics are compared to evaluate the error level. The evaluation is performed using Equation (2), where e represents the error value, $k_{mag}$ and $k_{phase}$ are the weighting factors for the magnitude and phase errors, respectively, and $Z_{mag}$ and $Z_{phase}$ represent the errors in the estimated impedance magnitude and phase, respectively.
(2)$$e=k_{mag}Z_{mag}+k_{phase}Z_{phase}$$(3)$pbest_{id}^{k}$ and $gbest_{id}^{k}$ update phase: In this phase, the evaluation values of each particle, determined by the fitness function are compared to the previously recorded best evaluation value of $pbest_{id}^{k}$. If the current evaluation value is lower than the previous best evaluation value, the parameter values are updated and assigned as the new $pbest_{id}^{k}$. Subsequently, among all the updated $pbest_{id}^{k}$ values, the value with the best evaluation result of $gbest_{id}^{k}$ is compared with the previous best evaluation result within the swarm. If the new evaluation result is superior, the derived parameters are updated. Once all the updates are completed, the algorithm checks whether the maximum number of iterations has been reached or if the evaluation result satisfies the termination criterion. If either condition is satisfied, the PSO algorithm terminates. This process ensures that the particles continuously update their personal best positions and velocities, while also considering the global best position within the swarm.(4)$x_{id}^{k+1}$ calculation phase: In this phase, when the maximum number of iterations is not exceeded and the evaluation result does not satisfy the termination criterion, new parameter values ($x_{id}$) are assigned for the next optimization operation. The new parameter values are determined according to the PSO algorithm, as shown in Equation (3). The velocity change ($v_{id}$), representing the direction and magnitude of the particle movement, is added to the current $x_{id}$ value, resulting in the derivation of the new $x_{id}$ value, as shown in Equation (4). This calculation phase ensures that each particle updates its position based on its previous position and the velocity change determined by the PSO algorithm, allowing for continuous exploration and refinement of the parameter space in the search for an optimal solution.
(3)$$V_{id}^{k+1}=wV_{id}^{k}+c_{1}r_{i}^{k}\left(pbest_{id}^{k}-x_{id}^{k}\right)+c_{2}r_{2}^{k}(gbest_{d}^{k}-x_{id}^{k})$$
(4)$$x_{id}^{k+1}=x_{id}^{k}+V_{id}^{k+1}$$

### 3.2. Proposed Method

As shown in Figure 5, The process of deriving the parameters to simulate multiple resonance characteristics and rapidly changing electrical impedance characteristics using multiple electrical RLC components is as follows:(1)Initialization phase: the necessary variables are initialized, and initial values are set for the PSO algorithm.(2)Multiple-component parameter derivation is through the PSO process, as shown in Figure 4, and the electrical parameters of the entire set of RLC components are derived, as shown in Figure 3.(3)Selection of restricted component range: component parameters that are physically difficult to implement are selected as the basis for defining the restricted range.(4)Selection of feasible components considering manufacturability: based on the upper and lower limits of the impedance determined in step 3, the RLC component parameters that exceed the impedance thresholds are selected.(5)PSO reiteration using the modified equivalent circuit: using the modified equivalent circuit from step 4, the components that were not selected are excluded, and the PSO derivation process is performed, as shown in Equations (3) and (4), to derive the electrical equivalent circuit parameters.

## 4. Results

The results of the equivalent circuit parameter estimation for single, double, and multiple resonant sonar sensors were compared to validate the accuracy of the proposed equivalent circuit and parameter estimation algorithm. The average error rates between the measured impedance magnitude and phase values (using Equations (5) and (6)) and the estimated impedance characteristics were utilized to compare the accuracy of the impedance characteristics. Here, $Z_{mag\_error\_avg}$ and $Z_{ph\_error\_avg}$, respectively, represent the average error rates of the impedance magnitude and phase. Further, $Z_{mag\_real}$, $Z_{mag\_est}$, $Z_{ph\_real}$, and $Z_{ph\_est}$ denote the measured and estimated values of the impedance magnitude and phase, respectively.

In this paper, the results of parameter derivation for circuit modeling are presented by comparing the conventional equivalent circuit and PSO algorithm-based parameter derivation with the proposed method using an equivalent circuit and the PSO algorithm. The results of the proposed algorithm for the parameters obtained in procedure (3) of Section 4.2 are denoted by “·” for electrical shorts and “X” for electrical opens in the table.
(5)$$Z_{mag\_error\_avg}\left[\%\right]=\frac{1}{n}\sum_{0}^{n}\sqrt{{\left(Z_{mag\_real}-Z_{mag\_est}\right)}^{2}}/\left(\frac{1}{n}\sum_{0}^{n}Z_{mag\_real}\right)\times 100$$
(6)$$Z_{ph\_error\_avg}\left[\%\right]=\frac{1}{n}\sum_{0}^{n}\sqrt{{\left(Z_{ph_{real}}-Z_{ph_{est}}\right)}^{2}}/\left(\frac{1}{n}\sum_{0}^{n}Z_{ph_{real}}\right)\times 100$$

### 4.1. Single Resonance Characteristic Results

As shown in Figure 6a, for a sonar sensor with a single resonant frequency within the operating frequency range, a conventional equivalent circuit with an RLC branch was constructed to simulate the single resonant mode in the BVD model. The equivalent circuit parameters were derived by utilizing the PSO algorithm, illustrated in Figure 4, in which the measured impedance data are represented by the red line, and the characteristics of the equivalent circuit are denoted by the line of crosses in Figure 6b. The values obtained for the four parameters of the equivalent circuit are presented in Table 3.

To compare the proposed equivalent circuit and derivation approach for a single-resonant sonar sensor, the results of deriving the equivalent circuit using the PSO algorithm shown in Figure 5 are summarized in Figure 7. Specifically, Figure 7a,b show the configurations of the equivalent circuits with 54 RLC components [EA] each. After excluding components with excessively large or small parameter values, which may appear electrically open or short, the equivalent circuit was sorted, as shown in Figure 7b. The parameter values for each component constituting the equivalent circuit are listed in Table 4 and Table 5. The component arrangement is organized in the order of subscripts from the top-left RLC component of $Z_{1}$ in the table, which corresponds to the 27 top-left RLC components, as shown in Figure 7a. The electrical characteristics of these components are shown in Figure 7c, wherein the blue and black lines represent the characteristic results. 

### 4.2. Dual Resonance Characteristic Results

Following the same procedure as that used for the comparison and validation of the equivalent circuit for single-resonant sonar sensors, we also compared the precision of the equivalent circuit results for sonar sensors with double resonant frequencies. The results of PSO obtained in Figure 4 were utilized to derive the equivalent circuit of a dual-resonant SONAR sensor, as presented in Figure 8 and Table 6 in the paper. 

The results of the parameter estimation using PSO are presented in Table 7 and Table 8. The characteristics of the equivalent circuit based on the parameters listed in Table 7 and Table 8 are shown in Figure 9, wherein the reference sensor characteristics are in red and the equivalent circuit characteristics are in blue.

### 4.3. Multiple Resonance Characteristic Results

To verify the feasibility of simulating multiple resonant characteristics and rapid impedance changes with varying frequencies, we compared the existing and proposed approaches based on the measured data for sensors with multiple resonant frequencies and characteristics exhibiting rapid variations.

Owing to the phase characteristics of the multi-resonant sonar sensor, an additional parallel $L_{1}$ was included in the existing BVD model to construct an equivalent circuit. The results of the parameter estimation using PSO are presented in Table 9. The characteristics of the equivalent circuit based on the parameters listed in Table 9 are shown in Figure 10, wherein the reference sensor characteristics are in red and the equivalent circuit characteristics are in blue.

To further compare and validate the proposed approach, we employed the PSO algorithm to derive the equivalent circuit and parameters, as shown in Figure 11. The results are presented in Table 10 and Table 11. The electrical impedance characteristics of each equivalent circuit in Figure 11c are represented by blue and black lines, as aforementioned.

Table 12 lists the average error rates relative to the reference impedance characteristics when different equivalent circuits were employed for each type of sensor. “Conventional” refers to the approach of deriving an equivalent circuit for each sonar sensor by modifying it based on the number of resonant frequencies. The terms “Before sorting” and “After sorting” define the stages in the process of utilizing the proposed 54 [EA] components-based equivalent circuit, where “Before” represents the initial result derived through the PSO algorithm, and “After” denotes the exclusion of certain components that are physically unsuitable.

Based on the average error rates for each equivalent circuit, the existing approaches were shown to have limitations in accurately estimating the impedance characteristics when the impedance exhibited rapid variations at the resonant points and frequencies, with maximum error rates of 5.64 and 12.34%, respectively. However, the proposed approach utilizing multiple components for the equivalent circuit and optimization algorithm demonstrated a relatively accurate simulation of the impedance characteristics, with maximum error rates of 1.43 and 2.52%, respectively, achieving average error rates within 3%. This validates the capability of accurately simulating the impedance characteristics and confirms that the proposed equivalent circuit and algorithm enable the derivation of the equivalent circuit of the sensor, regardless of the number of resonant frequencies.

## 5. Conclusions

In existing sonar sensor equivalent circuits, the circuits are modified based on the resonant characteristics of the sensor, and the limitations of the number of passive components within the circuit make it difficult to accurately reflect the impedance characteristics and rapidly changing impedance characteristics at adjacent resonant frequencies. To overcome these challenges, we developed an electrical equivalent circuit for sonar sensors based on their impedance characteristics and devised an estimation approach to derive the equivalent circuit.

To validate the proposed approach, we compared the average errors between the measured reference values and the estimated results of the equivalent circuit parameters for sensors with single, dual, and multiple resonances. The comparison results showed that, when using the conventional approach, the maximum average errors were 5.64 and 12.34% for the impedance magnitude and phase, respectively. However, when utilizing the proposed approach, the maximum errors were reduced to 0.8 and 2.58% for the impedance magnitude and phase, respectively, demonstrating higher precision compared to the conventional approach. Moreover, the proposed approach maintained high precision with average error rates below 1%, even with variations in the number of resonances and in the impedance characteristics of the sensor.

Furthermore, recent sonar sensors are not only operated at fixed frequencies but over a wide frequency range to enhance detection performance. Impedance matching is essential for overall system efficiency when operating a sensor across a wide frequency range. It is crucial to reflect the frequency characteristics of the load accurately when designing an impedance-matching circuit. The precision of the derived equivalent circuit, as previously described, is expected to enable the accurate reflection of load characteristics when designing amplifiers for sonar operation.

## Figures and Tables

**Figure 1 sensors-23-06636-f001:**
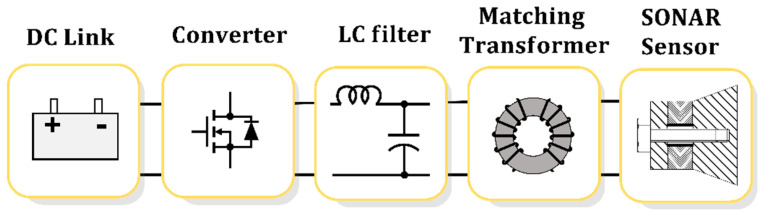
Sonar transducer power system.

**Figure 2 sensors-23-06636-f002:**
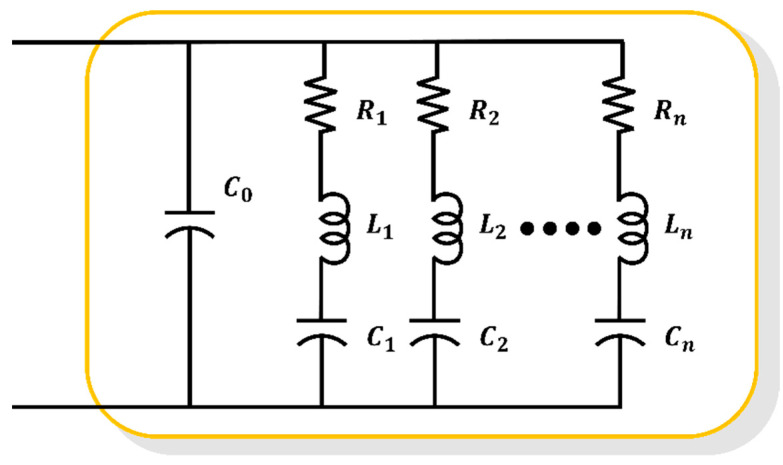
BVD equivalent circuit.

**Figure 3 sensors-23-06636-f003:**
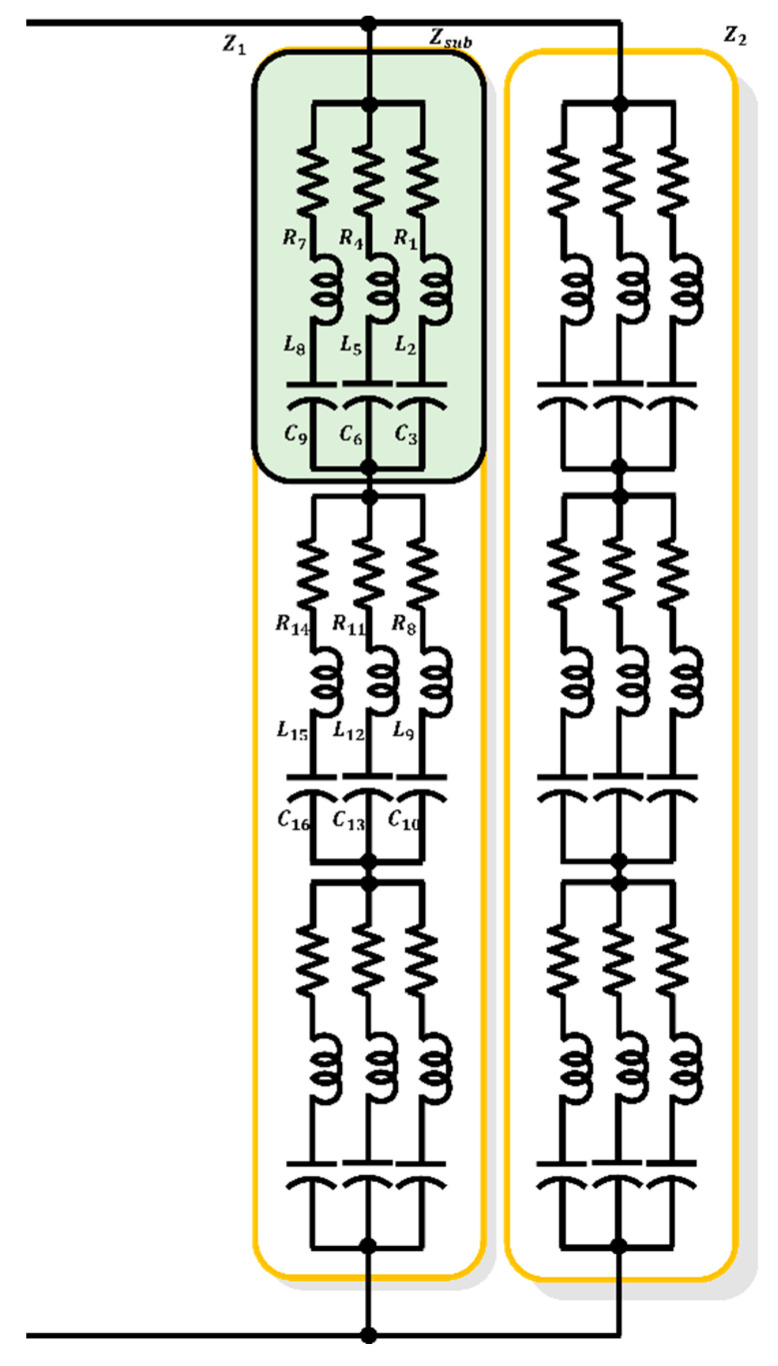
Proposed equivalent circuits, high-degree model.

**Figure 4 sensors-23-06636-f004:**
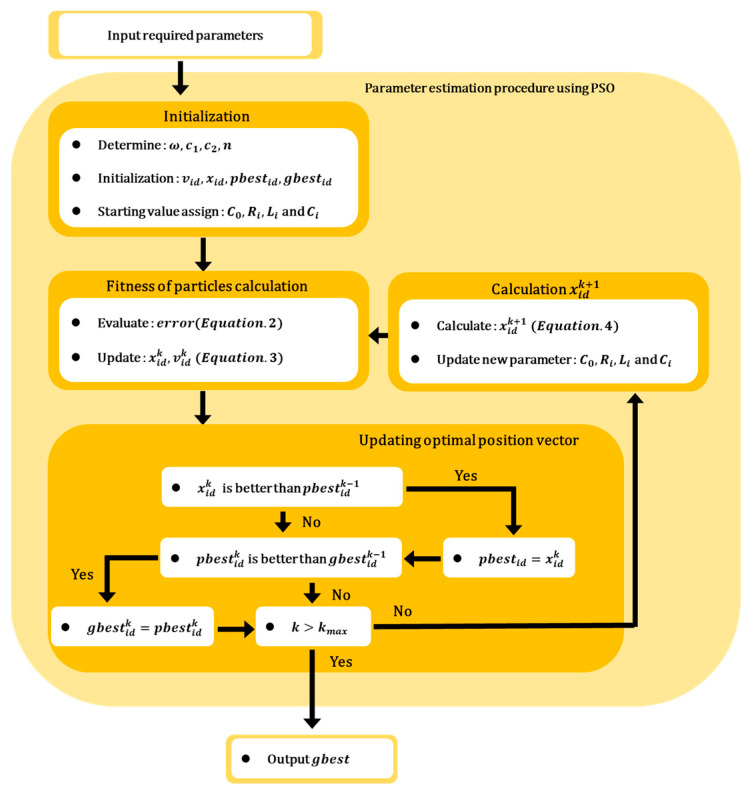
Conventional procedure for parameter estimation using PSO.

**Figure 5 sensors-23-06636-f005:**
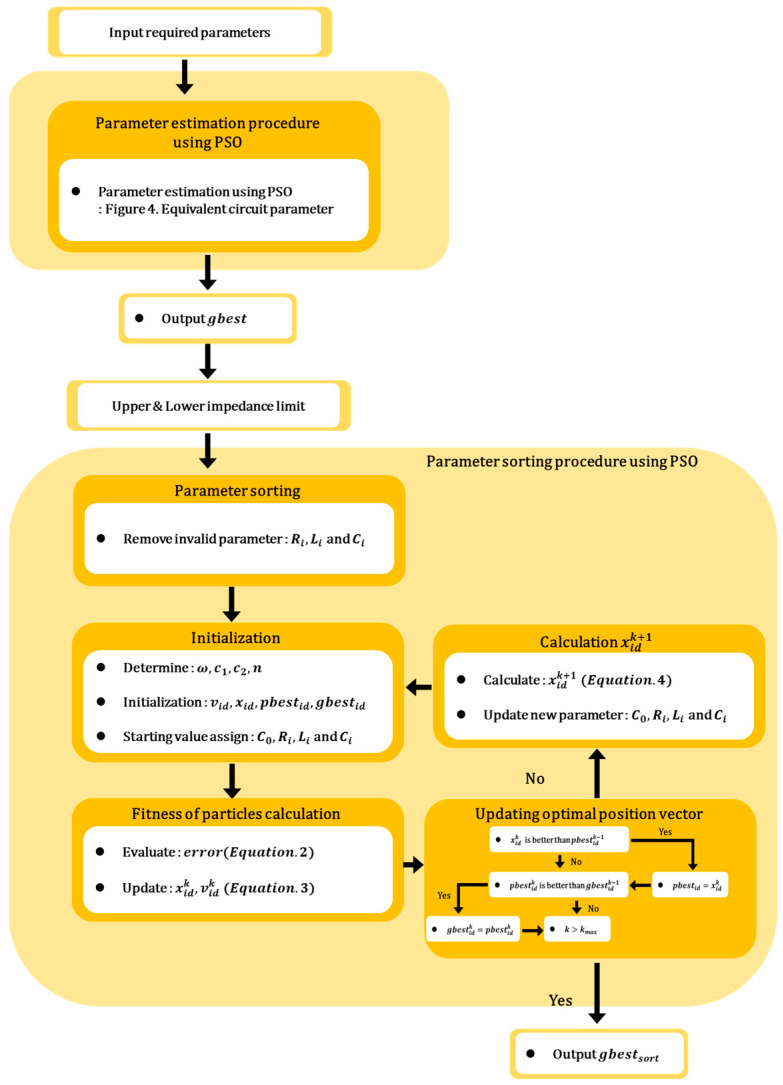
Proposed procedure of PSO algorithm for parameter estimation of high-degree electrical equivalent circuit.

**Figure 6 sensors-23-06636-f006:**
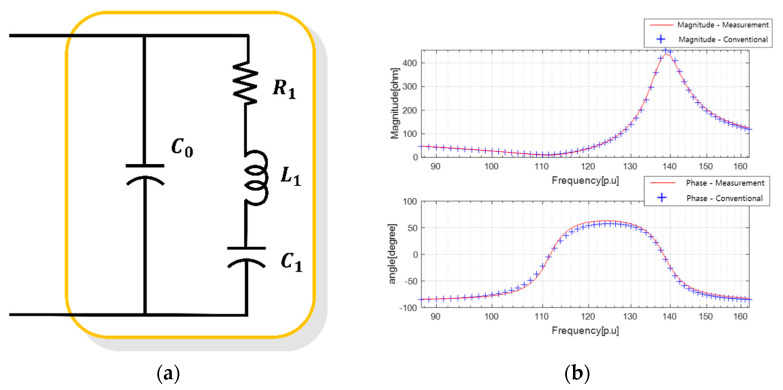
Results of deriving the equivalent circuit of a single resonant sensor using the conventional method: (**a**) conventional equivalent circuit; and (**b**) electrical impedance characteristics.

**Figure 7 sensors-23-06636-f007:**
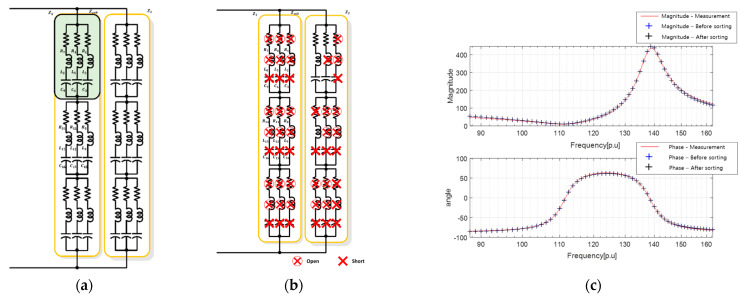
Results of deriving the equivalent circuit of a single resonant sensor using the proposed method: (**a**) high-degree equivalent circuit; (**b**) equivalent circuit after sorting unnecessary elements; (**c**) electrical impedance characteristics.

**Figure 8 sensors-23-06636-f008:**
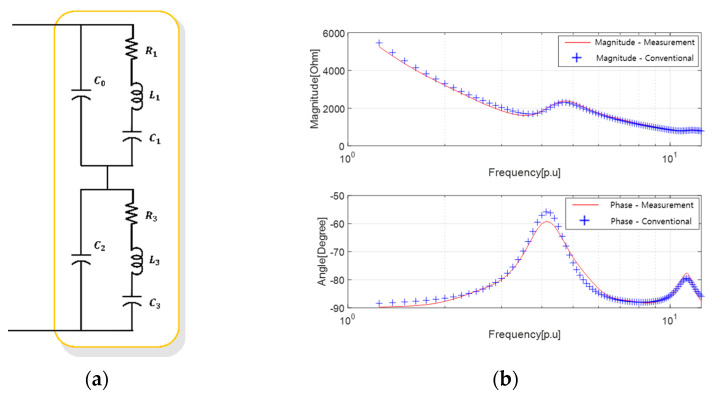
Results of deriving the equivalent circuit of the dual resonance sensor using the conventional method: (**a**) equivalent circuit; and (**b**) electrical equivalent characteristics.

**Figure 9 sensors-23-06636-f009:**
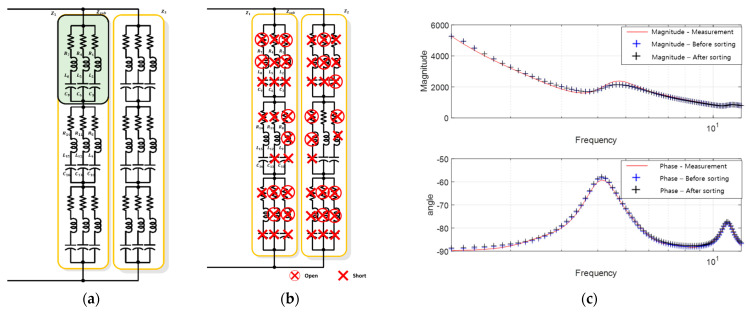
Results of deriving the equivalent circuit of a single resonant sensor using the proposed method: (**a**) high-degree equivalent circuit; (**b**) equivalent circuit after sorting unnecessary elements; and (**c**) electrical impedance characteristics.

**Figure 10 sensors-23-06636-f010:**
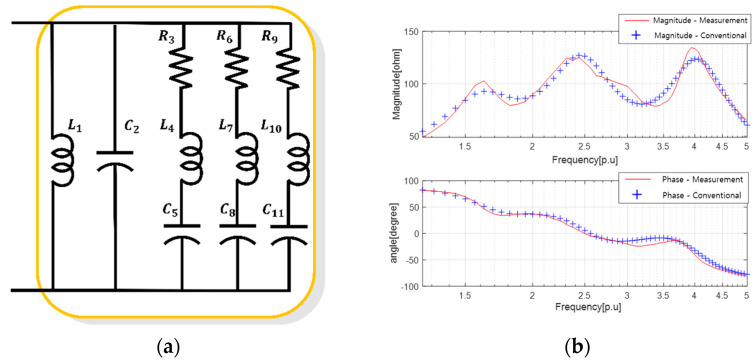
Results of deriving the equivalent circuit of the multiple resonance sensor using the conventional method: (**a**) equivalent circuit; and (**b**) electrical equivalent characteristics.

**Figure 11 sensors-23-06636-f011:**
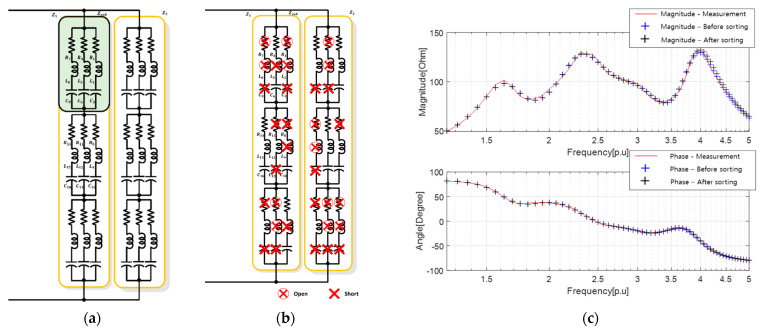
Results of deriving the equivalent circuit of the multiple resonant sensor using the proposed method: (**a**) high-degree equivalent circuit; (**b**) equivalent circuit after sorting unnecessary elements; and (**c**) electrical impedance characteristics.

**Table 1 sensors-23-06636-t001:** Characteristic of each type of parameter estimation algorithm.

Algorithm Type	Characteristic
Least square	The optimal solution is derived in the direction that minimizes the total sum of errors between the reference and estimated values.
Genetic algorithm	The optimal solution is found within a group, and the search is repeated based on the selected solution to derive the optimal solution.
PSO algorithm	Individual particles and clusters exchange error information based on one reference value to derive the optimal solution.

**Table 2 sensors-23-06636-t002:** Variable definitions for Equations (2)–(4).

Variables Name	Definition
$c_{1},\;c_{2}$	Acceleration constants
d	Number of swarm
n	Total number of parameters to derive PSO results
$r_{1},\;r_{2}$	Uniformly distributed random numbers
ω	Inertia weight factor
i	Number of particles
$p_{num}$	Total number of particles for the swarm
$V_{id}^{t}$	Present velocity vector of the swarm
$V_{id}^{t+1}$	Next velocity vector of the particle
$x_{id}^{k}$	Present position vector of the swarm
$x_{id}^{k+1}$	Next position vector of the particle
$gbest_{d}$	Optimal position vector of the swarm
$pbest_{d}$	Optimal position vector of the particle

**Table 3 sensors-23-06636-t003:** Results of deriving equivalent circuit parameters of a single resonant sonar sensor using the conventional method.

Parameter Name	Value	Parameter Name	Value
$C_{0}$	19.91 nF	$L_{1}$	287.58 μH
$R_{1}$	11.57 Ω	$C_{1}$	11.15 nF

**Table 4 sensors-23-06636-t004:** Results of deriving equivalent circuit parameters of a single resonant sonar sensor using the proposed method before sorting.

	R [Ω]	L [mH]	C [nF]	R [Ω]	L [mH]	C [nF]	R [Ω]	L [mH]	C [nF]
$Z_{1}$	1.30 × 10^31^	2.49 × 10^38^	4.13 × 10^35^	1.34 × 10^21^	6.27 × 10^32^	7.34 × 10^32^	2.51 × 10^30^	9.87 × 10^36^	1.37 × 10^35^
	3.09 × 10^34^	1.35 × 10^28^	6.33 × 10^33^	7.05 × 10^40^	2.08 × 10^34^	4.55 × 10^41^	3.00 × 10^23^	1.45 × 10^32^	1.31 × 10^33^
	2.43 × 10^33^	8.11 × 10^39^	3.75 × 10^38^	8.72 × 10^27^	2.37 × 10^38^	3.74 × 10^34^	2.30 × 10^28^	2.59 × 10^38^	9.62 × 10^32^
$Z_{2}$	3.22 × 10^31^	2.86 × 10^37^	7.54 × 10^38^	6.76 × 10^26^	1.97 × 10^34^	1.68 × 10^29^	0.0013	0.00013	7.14 × 10^31^
	2.99	7.25 × 10^−6^	17.1	1.81 × 10^31^	6.38 × 10^36^	1.45 × 10^34^	4.14 × 10^29^	2.29 × 10^40^	2.04 × 10^33^
	12.93	0.37	8.10	0.0025	0.018	8.30 × 10^27^	4.76 × 10^32^	1.16 × 10^31^	3.12 × 10^37^

**Table 5 sensors-23-06636-t005:** Results of deriving equivalent circuit parameters of a single resonant sonar sensor using the proposed method after sorting.

	R [Ω]	L [mH]	C [nF]	R [Ω]	L [mH]	C [nF]	R [Ω]	L [mH]	C [nF]
$Z_{1}$	X	X	·	X	X	·	X	X	·
	X	X	·	X	X	·	X	X	·
	X	X	·	X	X	·	X	X	·
$Z_{2}$	X	X	·	X	X	·	·	·	·
	2.99	·	17.1	X	X	·	X	X	·
	12.93	0.37	8.10	·	0.02	·	X	X	·

**Table 6 sensors-23-06636-t006:** Results of deriving equivalent circuit parameters of the dual resonance sonar sensor using the conventional method.

Parameter Name	Value	Parameter Name	Value
$C_{0}$	22 nF	$C_{2}$	390 nF
$R_{1}$	3.3 kΩ	$R_{3}$	12 Ω
$L_{1}$	425 mH	$L_{3}$	1.22 mH
$C_{1}$	5.6 nF	$C_{3}$	680 nF

**Table 7 sensors-23-06636-t007:** Results of deriving equivalent circuit parameters of the dual resonant sonar sensor using the proposed method before sorting.

	R [Ω]	L [mH]	C [nF]	R [Ω]	L [mH]	C [nF]	R [Ω]	L [mH]	C [nF]
$Z_{1}$	6.78 × 10^34^	8.39 × 10^28^	7.80 × 10^30^	0.00012	0.019	1813.33	0.71	1.19 × 10^−6^	3637.09
	1.37	8.47 × 10^−7^	15,210.46	8.18 × 10^−5^	9.64	2.59 × 10^34^	8.49 × 10^38^	1.60 × 10^35^	6.86 × 10^32^
	4.38 × 10^36^	7.73 × 10^34^	1.39 × 10^30^	64.89	9.88	595.98	0.00023	7.59	1.16 × 10^32^
$Z_{2}$	6.23	0.69	1910.65	0.00016	5.56	46,620.45	2.39 × 10^37^	3.95 × 10^35^	1.23 × 10^41^
	5.62 × 10^29^	2.31 × 10^36^	7.21 × 10^40^	64.78	21.10	525.72	9.72 × 10^37^	1.16 × 10^46^	1.42 × 10^33^
	0.81	1.34	4.75 × 10^8^	4.64 × 10^45^	3.51 × 10^34^	3.49 × 10^29^	2.12 × 10^−5^	0.35	1.03 × 10^44^

**Table 8 sensors-23-06636-t008:** Results of deriving equivalent circuit parameters of the dual resonant sonar sensor using the proposed method after sorting.

	R [Ω]	L [mH]	C [nF]	R [Ω]	L [mH]	C [nF]	R [Ω]	L [mH]	C [nF]
$Z_{1}$	X	X	·	X	X	·	X	X	·
	·	·	·	·	9.64	·	X	X	·
	X	X	·	·	9.88	595.98	·	7.59	·
$Z_{2}$	·	·	X	X	·	46,620.45	X	X	·
	X	X	·	21.1	526	525.72	X	X	·
	·	·	·	X	X	·	·	·	·

**Table 9 sensors-23-06636-t009:** Results of deriving equivalent circuit parameters of the multiple resonance sonar sensor using the conventional method.

Parameter Name	Value	Parameter Name	Value
$L_{1}$	5.356 mH	$L_{7}$	11.844 mH
$C_{2}$	806.5 nF	$C_{8}$	318.989 nF
$R_{3}$	10 Ω	$R_{9}$	129.631 Ω
$L_{4}$	6.344 mH	$L_{10}$	29.999 mH
$C_{5}$	149.46 nF	$C_{11}$	380.17 nF
$R_{6}$	90.877 Ω	-	-

**Table 10 sensors-23-06636-t010:** Results of deriving equivalent circuit parameters of the multiple resonant sonar sensor using the proposed method before sorting.

	R [Ω]	L [mH]	C [nF]	R [Ω]	L [mH]	C [nF]	R [Ω]	L [mH]	C [nF]
$Z_{1}$	6.78 × 10^34^	8.39 × 10^28^	7.80 × 10^30^	0.00012	0.019	1813.33	0.709	1.19 × 10^−6^	3637.09
	1.37	8.47 × 10^−7^	1.52 × 10^4^	8.18 × 10^−5^	9.64	2.59 × 10^34^	8.49 × 10^38^	1.60 × 10^35^	6.86 × 10^32^
	4.38 × 10^36^	7.73 × 10^34^	1.39 × 10^30^	64.89	9.88	595.98	0.00023	7.59	1.16 × 10^32^
$Z_{2}$	6.23	0.69	1.91 × 10^3^	0.00016	5.56	46,620.45	2.39 × 10^37^	3.95 × 10^35^	1.23 × 10^41^
	5.62 × 10^29^	2.31 × 10^36^	7.21 × 10^40^	64.78	21.1	525.72	9.72 × 10^37^	1.16 × 10^46^	1.42 × 10^33^
	0.81	1.34	4.75 × 10^8^	4.64 × 10^45^	3.51 × 10^34^	3.49 × 10^29^	2.12 × 10^−5^	0.35	1.03 × 10^44^

**Table 11 sensors-23-06636-t011:** Results of deriving equivalent circuit parameters of the multiple resonant sonar sensor using the proposed method after sorting.

	R [Ω]	L [mH]	C [nF]	R [Ω]	L [mH]	C [nF]	R [Ω]	L [mH]	C [nF]
$Z_{1}$	X	X	·	·	·	1813.33	0.71	·	3637.09
	1.37	Short	15,210.46	·	9.64	·	X	X	·
	X	X	·	64.89	9.88	595.98	·	7.59	·
$Z_{2}$	6.23	0.69	1910.65	·	5.56	46,620.45	X	X	·
	X	X	·	64.78	21.1	525.72	X	X	·
	0.81	1.34	·	X	X	·	·	0.35	·

**Table 12 sensors-23-06636-t012:** Average error rate according to the equivalent circuit by sonar sensor type.

	Equivalent Type	Impedance Magnitude Error Ratio [%]			Impedance Phase Error Ratio [%]		
Sensor Characteristic		Conventional	Before Sorting	After Sorting	Conventional	Before Sorting	After Sorting
Single resonant sensor		0.063	0.055	0.055	0.11	0.031	0.031
Dual resonant sensor		1.31	1.43	1.51	3.93	0.061	0.078
Multi-resonant sensor		5.64	0.018	0.82	12.34	0.62	2.58

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
