# Peer review of "Estimation Method of an Electrical Equivalent Circuit for Sonar Transducer Impedance Characteristic of Multiple Resonance"

_sensors, 2023, doi:10.3390/s23146636_

Round 1

Reviewer 1 Report

This paper proposed a high-degree equivalent electrical circuit model composed of 54 RLC components, the particle swarm optimization(PSO) algorithm is used to optimize the model. and it allows for the selection of different equivalent circuits for each resonance frequency. The proposed model can overcome the limitations of the sensor characteristic simulation due to the constraints on the number of RLC components. The article is innovative enough, but there are some questions and suggestions as follows:

1.     Is “any” of the article titles scientific? Does the proposed equivalent electrical circuit have a maximum and a minimum frequency in application, and what are its limitations?

2.     In this paper, the proposed high-degree model composed of 54 RLC components which depicted in Figure 4. But why 54 RLC components? Is the more RLC components the better?

3.     The proposed equivalent electrical circuit is optimized by PSO algorithm to make it as close to the measurement as possible, but there are many algorithms that can achieve the optimization, so it is recommended to investigate other optimization algorithms.

4.     In addition, the traditional equivalent circuit is introduced at the beginning of the second paragraph, which is lengthy and should belong to the introduction.

      5. Finally, the graphs and tables you draw should be modified. For example, Fig.        7(b), Fig. 8(c), Fig. 9(b), Fig. 10(c)Fig. 11(b) and Fig. 12(c), these figures              have two images but only one legend. And the R, L, C in the table 3, 4, 6, 7, 9            and 10 need to be distinguished.

The Quality of English Language is OK and has strong readability.

Reviewer 2 Report

In the manuscript titled Estimation Method of an Electrical Equivalent Circuit for any Sonar Transducer Impedance Characteristic, Jejin Jang et al performed an algorithm for deriving an equivalent circuit independent of resonance by employing multiple electrical components and particle swarm optimization (PSO).And using the Butterworth-Van Dyke (BVD) model, which is a method for deriving electrical equivalent circuits.

This study contains some interesting findings and are valuable for the understanding of sonar sensor equivalent circuits. However, Lack of innovation is the major flaw of the study

Therefore, MAJOR revision has to be done before this manuscript could be accepted for publication in the sensor.

Major comments

1. This could be an interesting study. Although it is adequately written, it offers no new information and no new slant on the topic.

2. Figure 12(b) is not clearly expressed, and the open and short are not easily identifiable.

3. It should be indicated why 54 RLC components were chosen to form a high-order model, and what are the advantages of 54 components?

4.When n frequencies, only 3 frequencies are given, which is not universal, not convincing, and should be generalized to more frequencies.

 The current manuscript needs to be polished by a native English speaker or a professional language editing service.
